# Strategy for Pediatric Patients with Relapsed or Refractory Anaplastic Lymphoma Kinase-Positive Anaplastic Large Cell Lymphoma: A Review

**DOI:** 10.3390/cancers15245733

**Published:** 2023-12-07

**Authors:** Kazuhiro Noguchi, Yasuhiro Ikawa

**Affiliations:** Department of Pediatrics, School of Medicine, Institute of Medical, Pharmaceutical and Health Sciences, Kanazawa University, 13-1 Takara-machi, Kanazawa 920-8641, Ishikawa, Japan; serori@staff.kanazawa-u.ac.jp

**Keywords:** anaplastic large cell lymphoma, ALK-positive, pediatric, target therapy, ALK inhibitor, brentuximab vedotin, vinblastine, hematopoietic stem cell transplantation, small cell variant, leukemic presentation

## Abstract

**Simple Summary:**

Anaplastic lymphoma kinase (ALK)-positive anaplastic large cell lymphoma (ALCL) is an aggressive T-cell lymphoma characterized by large T-cells with strong CD30 and ALK expression. Approximately 30% of patients treated with conventional chemotherapy experience a relapse or refractory disease and have a poor prognosis. Several risk factors associated with poor prognosis have been identified in pediatric ALK-positive ALCL, such as the morphological pattern of the small cell variant or lymphohistiocytic variant, leukemic presentation, the presence of minimal disseminated disease, or involvement of the central nervous system. Relapsed or refractory ALK-positive ALCL requires salvage therapy. Recently, targeted therapies such as ALK inhibitors and brentuximab vedotin have demonstrated dramatic responses in chemoresistant ALK-positive ALCL. Hematopoietic stem cell transplantation has also been reported to be an effective therapy. This article reviews pediatric ALK-positive ALCL, focusing on the risk factors associated with poor prognosis and treatment strategies for relapsed or refractory disease.

**Abstract:**

Anaplastic lymphoma kinase (ALK)-positive anaplastic large cell lymphoma (ALCL) is an aggressive T-cell lymphoma characterized by large T-cells with strong CD30 and ALK expression. Although conventional chemotherapy is effective in most patients, approximately 30% experience a relapse or refractory disease and have a poor prognosis. Several risk factors associated with poor prognosis have been identified in pediatric ALK-positive ALCL. These include morphological patterns with the small cell variant or lymphohistiocytic variant, leukemic presentation, the presence of minimal disseminated disease, or involvement of the central nervous system. Relapsed or refractory ALK-positive ALCL is often resistant to conventional chemotherapy; therefore, salvage therapy is required. In recent years, targeted therapies such as ALK inhibitors and brentuximab vedotin (BV) have been developed. ALK inhibitors block the continuous activation of ALK kinase, a driver mutation that leads to cell proliferation in ALK-positive ALCL. Additionally, BV is an antibody–drug conjugate that targets CD30-positive cells. Both ALK inhibitors and BV have displayed dramatic effects in chemoresistant ALK-positive ALCL. Weekly vinblastine treatment and hematopoietic stem cell transplantation have also been reported to be effective therapies. This article reviews pediatric ALK-positive ALCL, focusing on risk factors and treatment strategies for pediatric patients with relapsed or refractory ALK-positive ALCL.

## 1. Introduction

Anaplastic large cell lymphoma (ALCL) is an aggressive T-cell lymphoma characterized by large T-cells with strong CD30 expression [1]. Moreover, ALCL accounts for 10–15% of pediatric non-Hodgkin’s lymphoma cases and was first described in 1985 by Stein et al. [2]. ALCL is classified into two subtypes based on anaplastic lymphoma kinase (ALK) expression: ALK-positive ALCL and ALK-negative ALCL [3]. In pediatric patients, ALK-positive ALCL accounts for 80–100% of the cases, with a male predominance, whereas ALK-negative ALCL is more common in adults [1,2,4]. Although ALK-negative ALCL generally has a worse prognosis than ALK-positive ALCL in adults, pediatric patients with ALK-positive and ALK-negative ALCL have similar prognoses [4,5]. In more than 90% of patients with ALK-positive ALCL, a rearrangement of the *ALK* gene was detected. Specifically, nucleophosmin (*NPM*)-*ALK* was present in 80% of the cases, whereas tropomyosin 3 (*TPM3)-ALK* was present in 15% [6,7]. The *NPM-ALK* fusion gene results from a chromosomal translocation, t(2;5)(p23;q35), which brings together the *ALK* gene at 2p23 and the *NPM* gene at 5q35 [8]. Conventional chemotherapy is effective in treating ALK-positive ALCL and is associated with a favorable prognosis, with 5-year event-free survival rates (EFSs) ranging from 65 to 75% [9]. However, about 25–35% of ALK-positive ALCL patients experience relapsed or refractory disease. In addition, certain subtypes, such as “small cell variant” and “lymphohistiocytic variant,” have chemoresistant characteristics and poor prognosis, with a 5-year EFS of approximately 50% [10]. Relapsed or refractory ALK-positive ALCL is often resistant to conventional chemotherapy. Therefore, salvage therapy is required. In recent years, targeted therapies, such as ALK inhibitors and brentuximab vedotin (BV) have been developed, and they have demonstrated dramatic responses in chemoresistant ALK-positive ALCL [11,12,13]. Additionally, hematopoietic stem cell transplantation (HSCT) has been reported as an effective therapy for relapsed or refractory ALK-positive ALCL [14,15,16]. In this article, we review pediatric ALK-positive ALCL cases and discuss strategies for the treatment of pediatric patients with relapsed or refractory ALK-positive ALCL.

## 2. Clinical Features

ALK-positive ALCL typically manifests as a highly aggressive stage III to IV disease with systemic symptoms, especially high fever [1]. Extranodal lesions are frequently observed in ALK-positive ALCL (60%), such as those affecting the skin (19–21%), bone (17–19%), soft tissues (16–17%), bone marrow (11–12%), lungs (11–21%), and liver (8–14%) [4]. However, the involvement of the gut and central nervous system (CNS) is uncommon [4]. Bone marrow involvement is defined as the existence of ALCL cells in bone marrow determined through the analysis of the bone marrow smears. 

## 3. Oncogenic Mechanism

Rearrangements of the *ALK* gene have been detected in more than 90% of patients with ALK-positive ALCL. The expression of the wild-type ALK protein is strictly restricted to a few scattered cells in the CNS from birth [8]. Upon ligand binding, the wild-type ALK protein undergoes homodimerization, which activates tyrosine kinases in the intracellular tail of the ALK molecule. The NPM-ALK fusion protein consists of an NPM oligomerization domain and an intracytoplasmic ALK region, including the tyrosine kinase domain. The expression of the *NPM-ALK* fusion gene is controlled by the *NPM* promoter, resulting in the continuous production of the NPM-ALK protein [17]. Homodimerization of the NPM-ALK protein by the NPM oligomerization domain activates ALK tyrosine kinase, leading to cell proliferation [1]. Even in *TPM3* or tropomyosin-receptor kinase fused (*TFG*) genes, which are other partner genes of the *ALK* gene and include a dimerization domain, homodimerization of each fusion protein by dimerization domains contributes to the activation of ALK tyrosine kinase and oncogenic activity [1].

## 4. Risk Factors with Poor Prognosis

Several risk factors associated with poor prognosis have been identified in pediatric ALK-positive ALCL. These include morphological patterns with the small cell variant or lymphohistiocytic variant, leukemic presentation, MDD, and CNS involvement [18,19].

### 4.1. Morphological Pattern

The common type of ALK-positive ALCL is characterized by large tumor cells with horseshoe-shaped nuclei, abundant cytoplasm containing numerous vacuoles, and strong positive staining for ALK and CD30 [1]. The small cell variant is a subtype of ALK-positive ALCL characterized by the coexistence of mainly small tumor cells that stain negative or weakly positive for ALK and CD30, along with a minor population of large tumor cells that stain strongly positive for ALK and CD30 [1]. Additionally, the small cell variant accounts for 6% of all ALK-positive ALCL cases [20]. The small cell variant has been identified as a risk factor for relapsed or refractory diseases [18]. Small tumor cells in the small cell variant have been reported to be more resistant to chemotherapy than large tumor cells [21].

The lymphohistiocytic variant is characterized by numerous histiocytes surrounding CD30-positive large tumor cells [1]. Immunohistochemistry revealed that the histiocytes were negative for the nuclear proliferation marker Ki-67, indicating that they had assembled reactively rather than were undergoing neoplastic proliferation [1]. The lymphohistiocytic variant has been reported to be a risk factor for relapsed or refractory disease [18].

### 4.2. Leukemic Presentation

Leukemic presentation is defined as the presence of circulating ALCL cells in peripheral blood and is associated with an extremely poor prognosis. Leukemic presentation is extremely rare, accounting for less than 5% of all ALCL cases, and 75% of patients with leukemic presentation have a small cell variant histology [22]. Furthermore, most pediatric patients with leukemic presentation are associated with the *NPM-ALK* fusion gene. Diffuse lung infiltrates, respiratory distress, and pleural effusion are characteristic clinical features of leukemic presentation and have been reported in 50% of patients [22]. The diagnosis of ALK-positive ALCL with a leukemic presentation can be challenging because the condition mimics T-cell leukemia [22]. Identification of lymphoma cells with characteristic karyomorphism, such as “flower-cell” or “cerebriform cell,” will contribute to the diagnosis of leukemic presentation. (Figure 1) Moreover, when these characteristic cells present immunophenotypic findings as CD30- and ALK-positive, they can assist in the diagnosis of ALK-positive ALCL with a leukemic presentation [22]. Notably, lymphoma cells in peripheral blood often express myeloid-associated antigens, including CD13, CD11b, and lysozyme [22]. Whereas we encountered a patient with a leukemic presentation of a small cell variant whose circulating small tumor cells exhibited low expression of CD30 and ALK on immunohistochemistry and flow cytometry, with only one-tenth of *NPM-ALK* mRNA expression compared to large tumor cells [21]. This case highlights the importance of careful examination of cell morphology, even if CD30 and ALK expression in circulating tumor cells is low. Additionally, the lower percentage of tumor cells in the bone marrow than in the peripheral blood may contribute to the diagnosis of leukemic presentation [23]. As leukemic presentation has a very poor prognosis, an accurate diagnosis is crucial for selecting effective salvage therapies.

### 4.3. Minimal Disseminated Disease

Minimal disseminated disease (MDD) is defined as a minimal lesion detected in the peripheral blood and/or bone marrow using PCR to target the *ALK*-associated fusion gene. The *NPM-ALK* fusion gene is commonly targeted in many reports [18].

With regard to the PCR methods used to detect MDD, qualitative PCR (RT-PCR), RQ-PCR, and digital PCR (dPCR) have been reported. Damm-Welk et al. described that the sensitivity of RT-PCR was 10^−5^ NPM-ALK-positive cells per control cells [24]. In total, 48–61% of patients have a positive RT-PCR result in bone marrow, which indicates a poorer prognosis compared to those with a negative RT-PCR result [23,24,25]. They reported that, in positive and negative RT-PCR, the cumulative incidence of relapse was 50% and 15%, 5-year EFS was 38% and 82%, and 5-year OS was 60% and 86%, respectively [24]. The RQ-PCR method for *NPM-ALK* detects each copy number of *NPM-ALK* and internal control genes such as *ABL*. Then, NPM-ALK copy numbers are normalized using the copy numbers of internal control gene. The normalized copy numbers (NCNs) have been defined as *NPM-ALK* copy numbers per 10^4^ copies of *ABL* in several reports [24,26]. More than 10 NCNs are related to poor prognosis [24,26]. Damm-Welk et al. reported that in more than 10 NCNs and 10 or fewer NCNs in bone marrow, the cumulative incidence of relapse was 71% and 18%, 5-year EFS was 23% and 78%, and 5-year OS was 46% and 85%, respectively [24].In recent years, MDD detection using dPCR has been reported [27]. More than 30 NCNs have been linked poor prognosis [27,28]. A study by Damm-Welk et al. revealed that the 5-year progression-free survival (PFS) was 35%, 69%, and 74% for patients with more than 30 NCNs, 30 or fewer NCNs, and negative in bone marrow, respectively [27]. Although MDD quantification using the RQ-PCR method cannot be compared across different laboratories and international settings, the dPCR method demonstrates high inter-laboratory reproducibility [27].

### 4.4. CNS Involvement

In pediatric patients with ALCL, CNS involvement is rare, occurring in only 2.6% of cases [19]. Solitary ALCL localized in the CNS has been reported in only three cases [19]. Recently, CNS involvement was identified as a poor prognostic factor. Patients with CNS involvement have a worse prognosis than those with common ALK-positive ALCL, with a 5-year overall survival (OS) of 74% and 5-year EFS of 50% [19]. The lymphohistiocytic variant morphology and leukemic presentation have been reported as risk factors for CNS involvement. In particular, 36% of the patients with lymphohistiocytic variants developed CNS involvement [23].

## 5. Treatment

### 5.1. Treatment for Newly Diagnosed ALK-Positive ALCL

Anthracycline-based combination chemotherapy is the standard treatment for pediatric patients with newly diagnosed ALK-positive ALCL. Various national groups have reported the outcomes of treatment for patients with newly diagnosed ALCL (Table 1) [18,29,30,31,32,33,34,35]. The largest trial was the ALCL99 trial (NCT00006455), which included 420 pediatric and young adult patients with ALCL. Of these, 404 were ALK-positive and 16 were ALK-negative. The 5-year OS rate was 91% and the 5-year PFS was 72% [18]. Patients with the morphological pattern of the small cell variant or lymphohistiocytic variant had a poor prognosis, with a 10-year PFS of 50% compared to 79% for other morphologies (*p* < 0.0001) [18]. Patients with the small cell variant or lymphohistiocytic variant often exhibit a refractory course and require salvage therapies.

The Children’s Oncology Group (COG) has recently released the interim results of the ANHL12P1 trial, which examined the ALCL99 regimen with BV (arm BV) or crizotinib (arm crizotinib) for pediatric patients with newly diagnosed ALK-positive ALCL (NCT01979536). The arm BV included 68 patients with ALK-positive ALCL, with a 2-year OS of 97% and a 2-year EFS of 79% [36]. The addition of BV did not increase toxicity [25]. The crizotinib arm included 66 patients, with a 2-year OS of 95% and a 2-year EFS of 77% [37]. The addition of crizotinib increased thromboembolic events, which 13 out of 66 patients experienced [37]. In this study, the morphological subtypes were not analyzed.

The CHOP (cyclophosphamide, doxorubicin, vincristine, and prednisone) regimen is recommended in adult patients with newly diagnosed ALCL. For adult patients with limited stages, involved site radiation therapy is also generally recommended. A recent study by Horwitz et al. reported the results of the ECHELON-2 phase 3 trial of the A+CHP regimen, which is a BV with combination chemotherapy consisting of cyclophosphamide, doxorubicin, and prednisone (CHP) for adult patients with newly diagnosed CD30-positive peripheral T-cell lymphoma, including ALCL (NCT01777152) [38]. The study reported that the A+CHP regimen had a better median PFS compared to the CHOP regimen (48.2 months vs. 20.8 months, *p* = 0.011), while the risk of adverse events was equivalent in both regimens. Consequently, the A+CHP regimen is considered a more effective treatment option for adult patients, positioning the modality as the standard next-generation treatment [38].

### 5.2. Treatment for Relapsed or Refractory ALK-Positive ALCL

No established standard treatment is available for relapsed or refractory ALK-positive ALCL. However, several treatments have been reported to be effective, including weekly vinblastine, BV, and ALK inhibitors. In addition, nivolumab has also been reported as a potential new treatment option. After complete remission, HSCT has also been reported as a curative treatment for relapsed or refractory ALK-positive ALCL (Table 2).

#### 5.2.1. Weekly Vinblastine

A report by the French Society for Pediatric Oncology (SFOP) highlighted the effectiveness of weekly vinblastine treatment (6 mg/m^2^/week) in pediatric patients with relapsed or refractory ALK-positive ALCL. Of the 30 evaluable patients, 25 (83%) patients achieved a complete response (CR) after weekly vinblastine treatment [9]. Notably, seven patients remained in CR for more than 2 years after the end of vinblastine treatment without HSCT. This indicated that vinblastine treatment has the potential to be curative. Vinblastine re-administration was also effective in patients with relapse who terminated the vinblastine salvage therapy. Six of the seven patients with relapse achieved CR with vinblastine treatment [9]. The 5-year OS was 65% and the 5-year EFS was 30% [9]. However, the optimal duration of treatment remains unclear. Notably, as vinblastine has a low penetrance into the CNS, relapse in the CNS requires caution during vinblastine treatment [46]. The most common adverse event with weekly vinblastine administration was mild leukopenia (34%) [9,39]. The incidence of extrahematological toxicity was low.

#### 5.2.2. Brentuximab Vedotin

Brentuximab vedotin (BV) is an antibody–drug conjugate that targets CD30-positive cells. The CD30-antibody is linked to the microtubule-disrupting agent monomethyl auristatin E (MMAE) [11]. Locatelli et al. reported the results of a phase 1/2 study on BV in pediatric patients with relapsed or refractory ALCL (NCT01492088). The study included 17 pediatric patients with ALCL, of whom 12 were ALK-positive and 5 were ALK-negative. The overall response rate was 53%, with a CR rate of 41% (7 of the 17 patients). Notably, 9 of the 17 patients had undergone HSCT [11]. In adult patients with ALCL, Pro et al. reported the results of a phase 2 study of BV with relapsed or refractory ALCL (NCT00866047). The study included 57 adult patients, 16 of whom were ALK-positive, and 42 were ALK-negative. The overall response rate was 86%, with a CR rate of 57% (33 of the 58 patients) [12]. The 5-year OS rate of all patients treated with BV was 60%. This study also revealed that the 5-year PFS rates in patients with or without consolidative HSCT after BV treatment were 69% and 48%, respectively [13]. The 5-year OS rates were 75% and 81%, respectively [13]. Thus, consolidative HSCT improved the PFS in patients with CR who responded to BV treatment. Intriguingly, 8 of the 58 patients (14%) remained in sustained CR for more than 5 years without HSCT or other treatments. This result suggests that BV has the potential to cure relapsed or refractory ALCL [13]. Notably, as BV has low penetrance into the CNS, relapse in the CNS requires caution during BV treatment [46].

The most common adverse events in the 36 pediatric patients treated with BV were fever (44%), nausea (36%), and peripheral neuropathy (33%) [11]. Peripheral neuropathy with BV is caused by tubulin inhibition by MMAE and is dose-limiting and cumulative [47]. However, the aforementioned effects are typically transient and reversible. According to a report by Locatelli et al., 11 of 12 patients who developed peripheral neuropathy with BV recovered without sequela [11]. The most common grade 3 and higher adverse events were a decrease in neutrophil count (11%), increased gamma-glutamyl transpeptidase (6%), and fever (6%) [11]. Similar adverse events have been observed in adult and pediatric patients treated with BV. Among the 58 adult patients treated with BV, the most common adverse events were peripheral neuropathy (41%), nausea (40%), and fever (34%). The most common grade 3 and higher adverse events were decreased neutrophil count (21%), thrombocytopenia (14%), and peripheral neuropathy (12%) [12]. In adult patients, peripheral neuropathy is also the most transient and reversible, as 30 of 33 patients with BV-related peripheral neuropathy recovered [13]. Pancreatitis and progressive polyfocal leukoencephalopathy are rare, severe adverse events [48,49].

Repeated BV administration may lead to acquired drug resistance. Some studies have suggested that drug resistance is caused by resistance to MMAE, multidrug resistance protein 1 upregulation, and CD30 downregulation [50,51]. While some reports indicated that CD30 expression does not affect the efficacy of BV [47,52]. The contribution of CD30 downregulation in BV drug resistance remains controversial.

#### 5.2.3. ALK Inhibitors

Continuous activation of ALK kinase caused by the ALK fusion protein is a driver mutation in ALK-positive ALCL, leading to the continuous growth of cancer cells. Therefore, ALK inhibitors have been developed to prevent the proliferation of cancer cells. Moreover, ALK inhibitors have displayed a high overall response rate in relapsed and refractory ALK-positive ALCL. These inhibitors were initially developed for ALK-rearrangement non-small cell lung cancer (NSCLC) [53]. Currently, five ALK inhibitors have been developed: crizotinib, alectinib, ceritinib, brigatinib, and lorlatinib. They are classified into first to third generations, and only the first-generation ALK inhibitor crizotinib and second-generation ALK inhibitor alectinib have been approved for ALK-positive ALCL. Crizotinib was approved by the United States Food and Drug Administration (FDA) in 2021, and alectinib was approved by the Ministry of Health, Labour and Welfare of Japan in 2020. Although the second-generation ALK inhibitor ceritinib has not been approved for ALK-positive ALCL, several studies have indicated that ceritinib is effective for relapsed or refractory ALK-positive ALCL [42,54].

##### Crizotinib

Crizotinib is a first-generation ALK inhibitor approved for patients with relapsed or refractory ALK-positive ALCL in the United States of America in 2021. The COG reported the results of a clinical trial of crizotinib in pediatric patients with relapsed or refractory ALK-positive ALCL (NCT00939770). This study included 26 patients with ALK-positive ALCL. The overall response rate was 88%, with a CR rate of 81% (21 of the 26 patients) [43]. Eventually, 12 of 26 patients underwent HSCT. Currently, whether crizotinib has the potential to treat relapsed or refractory ALCL is unknown. Therefore, after achieving CR with crizotinib treatment, patients may undergo HSCT or continue receiving crizotinib as maintenance therapy. The most common grade 3 and higher adverse event was a decrease in the neutrophil count, which was reported in 33% of patients treated with crizotinib at a 165 mg/m^2^/dose and 70% of patients treated with a 280 mg/m^2^/dose [43]. Visual disturbances such as photopsia, blurred vision, and vitreous floaters are also common adverse events reported in adult patients, with an incidence rate of 40–60% [40,55]. These visual disturbances were mostly transient and occurred within the first week of initiating crizotinib treatment [55]. Adverse gastrointestinal events, including nausea (54%), diarrhea (42%), vomiting (41%), and constipation (29%) were also commonly reported in adult patients who received crizotinib [40].

When crizotinib is used as maintenance therapy, the optimal timing of crizotinib discontinuation remains unclear. Even if patients achieve and maintain CR for several years with crizotinib maintenance therapy, immediate relapse caused by crizotinib discontinuation has been reported in both pediatric and adult patients [56]. In this study, two patients who had been in CR for 45 and 27 months relapsed within 20 days after discontinuing crizotinib [56]. Notably, since crizotinib has low CNS penetrance, CNS relapse requires caution during crizotinib treatment [46,57]. Repeated crizotinib administration may lead to acquired drug resistance. Some studies on both ALK-positive ALCL and ALK-positive NSCLC have suggested that acquired resistance to crizotinib is linked to various mutations in ALK, including the L1196M *ALK* mutation, known as the gatekeeper mutation in the ALK kinase domain [58,59].

##### Alectinib

Alectinib is a second-generation ALK inhibitor that overcomes various crizotinib-resistant ALK mutations, including the L1196M gatekeeper mutation [60]. Fukano et al. reported the results of a phase 2 trial of alectinib in pediatric patients with relapsed or refractory ALK-positive ALCL (UMIN000016991). In this study, patients received oral doses of alectinib 300 mg twice daily, except for those weighing less than 35 kg, who received 150 mg twice daily. This study included 10 patients with ALK-positive ALCL. The overall response rate was 80%, with a CR rate of 60% (6 of the 10 patients) [61]. Eventually, 2 of the 10 patients underwent allogeneic HSCT in CR following alectinib treatment.

Alectinib is associated with a low incidence of adverse gastrointestinal events. Furthermore, gastrointestinal adverse events such as nausea, vomiting, and diarrhea have been reported in less than 15% of adult patients with NSCLC who received alectinib. The frequency of gastrointestinal adverse events of alectinib was lower than that of other ALK inhibitors [40]. The most common grade 3 or higher adverse event was a decrease in neutrophil count (20%) [61]. Currently, whether alectinib has the potential to treat relapsed or refractory ALCL is unknown. Therefore, after achieving CR with alectinib treatment, patients may undergo HSCT or continue to receive alectinib as maintenance therapy. Several patients have been reported to survive for over a year with continuous alectinib administration as well-tolerated maintenance therapy [41,61].

Although alectinib overcomes various crizotinib-resistant *ALK* mutations, continuous administration of alectinib induces acquired drug resistance caused by *ALK* mutation in patients with NSCLC and ALK-positive ALCL [62,63]. Additionally, the gatekeeper L1196M *ALK* mutation, which is usually overcome by alectinib, can also lead to alectinib resistance. An in vitro study demonstrated that echinoderm microtubule-associated protein-like 4 (*EML4*)*-ALK*-expressing cells that acquired L1196M *ALK* mutations were more susceptible to alectinib than crizotinib but less susceptible to alectinib than cells with wild-type ALK [62]. We have reported a pediatric patient with relapsed ALK-positive ALCL caused by an acquired L1196M *ALK* mutation during alectinib administration [63]. The most notable characteristic of alectinib is the high penetrance of the drug in the CNS, in contrast to crizotinib, vinblastine, and BV. A linear relationship exists between the concentration of alectinib in the cerebrospinal fluid and the unbound systemic concentration [40]. Alectinib has been reported to be effective in treating patients with relapsed or refractory CNS lesions caused by ALK-positive ALCL [21,64].

##### Ceritinib

Ceritinib is a second-generation ALK inhibitor. Although the drug has not yet been approved for ALK-positive ALCL, one study reported that three patients with relapsed or refractory ALK-positive ALCL were treated with ceritinib, of which two patients achieved CR and one patient achieved partial response [54]. Additionally, an independent report indicated that ceritinib treatment led to CR in a patient with relapsed and refractory ALK-positive ALCL [42]. In a previous study investigating the susceptibility of *EML4-ALK* expressing cells with various *ALK* mutations to ALK inhibitors, ceritinib proved effective against crizotinib- and alectinib-resistant *ALK* mutations such as I1171N and I1171S [62]. Currently, whether ceritinib has the potential to cure relapsed or refractory ALCL is unknown. Adverse gastrointestinal events, including nausea (82%), diarrhea (75%), vomiting (65%), and constipation (32%), are commonly reported in adult patients with NSCLC who receive ceritinib [65]. The most common grade 3 or higher adverse events reported were an increase in the levels of alanine aminotransferase (21%), aspartate aminotransferase (11%), and lipase (7%), along with an increase in diarrhea (7%) [65].

#### 5.2.4. Nivolumab

Nivolumab is an antibody that blocks the ligand activation of the programmed death-1 cell receptor (PD-1). The programmed death ligand 1(PD-L1), which is activated by the NPM-ALK fusion protein, leads to PD1 activation, resulting in immune system suppression and allowing the tumor to evade immune control. Several patients with relapsed or refractory ALK-positive ALCL have been reported to respond to nivolumab and achieve CR [44,45]. The NIVO-ALCL trial (NCT 03703050) is based on these findings and is ongoing to test nivolumab monotherapy as a consolidative immunotherapy or in case of progression after target therapies in patients with relapsed or refractory ALCL.

#### 5.2.5. Hematopoietic Stem Cell Transplantation

Hematopoietic stem cell transplantation (HSCT) can be a curative treatment if used in consolidation for relapsed or refractory ALK-positive ALCL. Autologous and allogeneic HSCT have also been previously reported.

##### Autologous HSCT

At first, the SFOP reported the effectiveness of HSCT in 15 pediatric patients with relapsed ALCL, including 14 autologous and 1 allogeneic HSCT [66]. The result demonstrated a 3-year DFS of 45%. Woessmann et al. analyzed 37 pediatric patients with relapsed ALCL who underwent reinduction chemotherapy followed by autologous HSCT. The results demonstrated a 5-year EFS of 59 ± 8% and a 5-year OS of 77 ± 7% after the first relapse [14]. This study also revealed that the timing of first relapse and CD3 expression in lymphoma cells significantly affected the outcomes of autologous HSCT. Patients who underwent autologous HSCT for relapse during the frontline therapy had a very poor prognosis, with an EFS of 0%. Furthermore, patients with CD3-positive ALCL who underwent autologous HSCT had a worse prognosis than those with CD3-negative ALCL (EFS:18% vs. 72%). Therefore, allogeneic HSCT is recommended for patients with relapse during the frontline therapy and CD3-positive ALCL [14]. Several studies have compared the effectiveness of autologous and allogeneic HSCT and found that allogeneic HSCT leads to better EFS rates than autologous HSCT [67]. Gross et al. reported a higher 5-year EFS for allogeneic HSCT (46%) compared to autologous HSCT (35%) [68]. Similarly, Fukano et al. reported that allogeneic HSCT showed better EFS than autologous HSCT: 5-year EFS 50% vs. 38% [40].

Moreover, Knorr et al. have recently reported the results of the ALCL-Relapse trial (NCT00317408): autologous HSCT was less effective for pediatric patients who experienced early relapse within one year after initial diagnosis [69]. In their study, autologous HSCT was less effective than allogeneic HSCT in patients with early relapse, regardless of CD3 positivity. The 5-year EFS for CD3-positive ALCL patients was 25% with autologous HSCT, compared to 65% with allogeneic HSCT. Similarly, the 5-year EFS for CD3-negative ALCL patients was 30% with autologous HSCT, compared to 78% with allogeneic HSCT. Given this, the authors concluded that autologous HSCT should not be undertaken as consolidation therapy for pediatric patients with relapsed or refractory ALCL [69].

##### Allogeneic HSCT

Strullu et al. reported the effectiveness of allogeneic HSCT in 34 pediatric patients with relapsed or refractory ALK-positive ALCL. The 5-year OS was 70% and the 5-year EFS was 58% [15]. In another study, the Berlin-Frankfurt-Munster group (BFM) reported the efficacy of allogeneic HSCT in 20 pediatric patients with relapsed or refractory ALK-positive ALCL. The 3-year EFS was 75% [16]. Allogeneic HSCT is also effective in patients with chemotherapy-resistant diseases, suggesting a graft-versus-ALCL effect [16]. As mentioned above, several studies comparing the effects of autologous and allogeneic HSCT discovered that allogeneic HSCT resulted in better EFS rates compared to autologous HSCT [67]. However, up-front allogeneic HSCT is not recommended for ALK-positive ALCL patients who achieve a partial or complete response following induction therapy [70]. The efficacy of up-front allogeneic HSCT has not been reported, even in patients with risk factors for poor prognosis, such as morphological patterns with small cell variant or lymphohistiocytic variant, leukemic presentation, MDD, and CNS involvement.

In conditioning regimens followed by allogeneic HSCT, the reduced-intensity conditioning (RIC) regimen resulted in better outcomes than the myeloablative conditioning (MAC) regimen [71]. Fukano et al. reported that the RIC regimen leads to better OS and EFS as compared to the MAC regimen: 5-year OS 100% vs. 48.5% and 5-year EFS 87.5% vs. 42.8% [71]. By reducing treatment-related mortality, the RIC regimen may improve allogeneic HSCT outcomes compared to the MAC regimen. The JPLSG-ALCL-RIC18 trial is ongoing in Japan to test the efficacy of the RIC regimen followed by allogeneic HSCT.

Patients with a leukemic presentation of refractory ALK-positive ALCL may require allogeneic HSCT. As ALK-positive ALCL with leukemic presentation is extremely rare, no consensus on the standard treatment is present [23]. Out of the 24 patients previously reported in the literature with leukemic presentation, including the information on their survival status, only 8 patients (33%) were alive [20,21,22,23,72,73,74,75,76,77,78,79]. The eight patients with leukemic presentation are summarized in Table 3 [20,21,22,23,72,74]. Of the eight patients, only two achieved a CR after frontline chemotherapy (cases 1 and 2). The patients survived without additional treatment, including HSCT. In contrast, the remaining six patients who did not achieve CR with frontline chemotherapy received salvage chemotherapy. Four out of six patients who achieved CR underwent HSCT as consolidation therapy, while the remaining two patients who achieved non-CR after salvage chemotherapy underwent HSCT as salvage therapy. All three patients who provided information about their conditioning regimen received total-body irradiation (TBI) as part of their treatment. Furthermore, all four patients reported that their donor source was allogeneic HSCT. Therefore, allogeneic HSCT with a TBI conditioning regimen may be necessary for patients with chemoresistant ALK-positive ALCL with a leukemic presentation.

An important point to note is that the high incidence of CNS relapse implies that CNS-directed therapy should not be omitted in patients with a leukemic presentation [23].

The optimal timing for discontinuing ALK inhibitors remains unclear when they are used as a bridging therapy for HSCT. In some reports, ALK inhibitors were terminated a day before conditioning therapy for HSCT [57,80]. However, a pediatric patient who terminated treatment with an ALK inhibitor a day before conditioning therapy relapsed 40 days after HSCT [63]. Similar to other treatment strategies, re-administration of ALK inhibitors as maintenance therapy following HSCT has been reported [57]. Establishing the optimal timing for discontinuation of ALK inhibitors before HSCT and determining the appropriate maintenance therapy after HSCT is warranted. In addition, involved site radiation therapy may be considered in refractory patients with focal disease [81].

#### 5.2.6. Treatment Strategy for Pediatric Patients with Relapsed or Refractory ALK-Positive ALCL

The standard therapy for pediatric patients with relapsed or refractory ALCL has not yet been firmly established. However, recent studies have shed light on the recommended treatment options. The result of the ALCL-Relapse trial (NCT00317408), the first prospective trial on pediatric patients with relapsed ALCL, can help to determine the best course of treatment. In the ALCL-Relapse trial, patients were classified into four risk groups based on the time of relapse and CD3 expression, and reinduction approaches were selected according to these four risk groups [69]. Patients with progression during frontline therapy were classified as very high risk. Patients with a CD3-positive relapse were classified as high risk. Patients with a CD3-negative relapse within one year after initial diagnosis were classified as intermediate risk. Patients with a CD3-negative relapse later than one year after initial diagnosis were classified as low risk. Patients with very high risk and high risk treated with allogeneic HSCT after reinduction chemotherapy showed good outcomes, with 5-year EFS of 41% and 62%, and 5-year OS of 59% and 73%, respectively, while patients with intermediate risk treated with autologous HSCT after carmustine-etoposide-cytarabine-melphalan showed poor outcome, with 5-year EFS of 30% and 5-year OS of 78%. This result indicates that autologous HSCT is not recommended for patients with early relapse within one year after initial diagnosis. Patients with low risk treated with weekly vinblastine for 24 months showed good prognosis, with 5-year EFS of 81% and 5-year OS of 90% [69]. Ultimately, allogeneic HSCT is recommended as standard consolidation therapy for pediatric patients with progression during frontline therapy or relapse of a CD3-positive ALCL. Furthermore, weekly vinblastine monotherapy for 24 months is recommended for pediatric patients with late relapse, later than 12 months after initial diagnosis, of a CD3-negative ALCL. Although there is no consensus of treatment for pediatric patients with early relapse of CD3-negative ALCL, within 12 months after initial diagnosis, allogeneic HSCT may be suitable for them and should be considered for future testing.

In conditioning regimens followed by allogeneic HSCT, a retrospective study revealed that the RIC regimen resulted in better outcomes than the MAC regime [71]. The JPLSG-ALCL-RIC18 trial is currently ongoing in Japan to evaluate the efficacy of the RIC regimen followed by allogeneic HSCT prospectively.

Although both ALK inhibitors and BV can induce CR in 40–80% of pediatric patients with relapsed or refractory ALCL, it remains unclear whether these drugs can offer cure without additional therapy [11,43,61]. Therefore, ALK inhibitors and BV should be administered as reinduction therapy before consolidation using allogeneic HSCT in all but patients with late relapse of a CD3-negative ALCL [69]. It should be noted that some of these therapies may be off-label for pediatric patients in certain countries.

As a strategy for patients with characteristic risk factors, we suggest treatment in patients with leukemic presentation and CNS involvement.

For patients with leukemic presentation showing chemoresistance, we suggest undergoing allogeneic HSCT with a TBI conditioning regimen, which has been reported to rescue patients [20,21,22].

For patients with CNS involvement, it is recommended to use treatment regimens including HD-MTX and/or HD-AraC as CNS-penetrating chemotherapy [19]. Additionally, cranial radiotherapy (18–24 Gy) has also been reported to be effective for ALCL patients with CNS involvement, especially those with intracerebral mass [19]. Recent studies have shown that alectinib, an ALK inhibitor that penetrates the CNS, is also effective in treating CNS involvement and may even replace cranial radiotherapy [21,82]. The necessity of cranial radiotherapy for ALCL patients with CNS involvement will require further investigation due to neurological sequelae.

## 6. Conclusions

ALK-positive ALCL typically responds well to conventional chemotherapy, as demonstrated in the ALCL99 trial. However, patients with risk factors, such as the small cell variant or lymphohistiocytic variant, leukemic presentation, MDD-positive status, or CNS involvement, may experience a refractory course, requiring salvage therapies, including weekly vinblastine, BV, ALK inhibitors, and HSCT. Especially for patients with CNS involvement, a CNS-penetrating drug such as alectinib, HD-MTX, or HD-AraC should be considered. In the future, we need a solution for acquired drug resistance in salvage therapies, including BV and ALK inhibitors, and the development of novel treatments for relapsed or refractory ALK-positive ALCL.

## Figures and Tables

**Figure 1 cancers-15-05733-f001:**
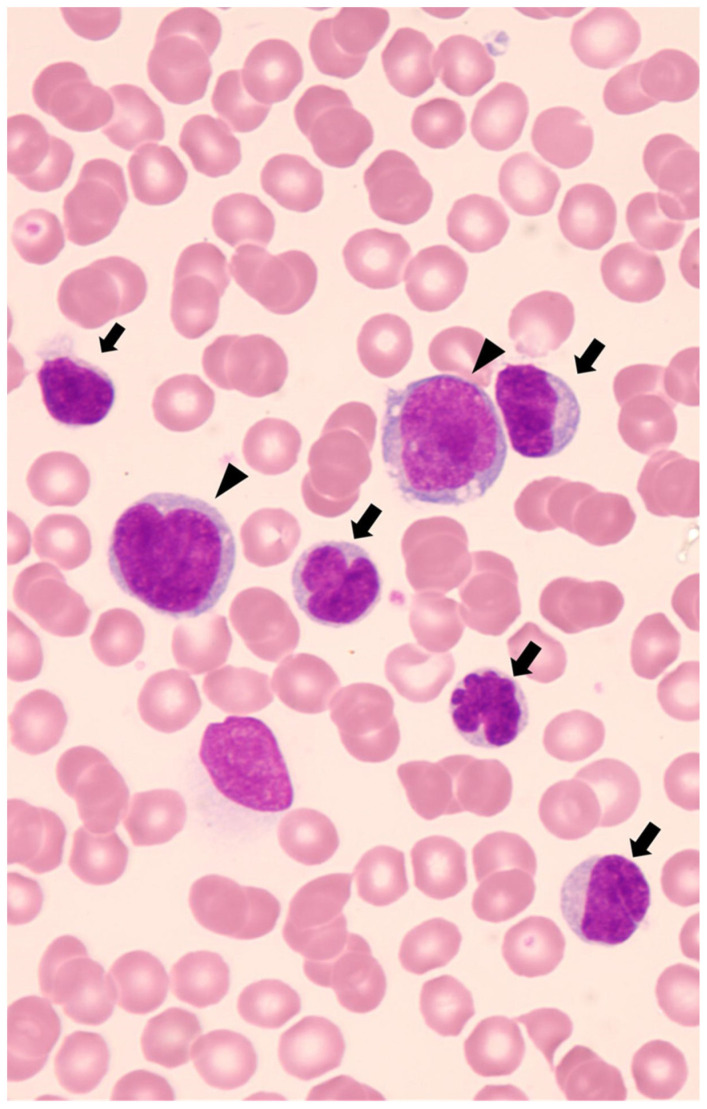
May-Giemsa-stained blood smears at the onset. Numerous small-sized lymphocytes with lobulated nuclei called “flower-cell” or “cerebriform cell” (arrow), and a few large-sized lymphocytes with basophilic and vacuolated cytoplasm (arrowhead) are displayed.

**Table 1 cancers-15-05733-t001:** Clinical trials for pediatric patients with ALK-positive ALCL.

Protocol	Study Group	Study Period	Treatment Strategy	Number of Patients	EFS (Year)	OS (Year)	Reference
HM89/91	SFOP	1988–1997	COPADM (CY, DXR, PSL, MTX, VCR) × 2 with maintenance treatment	82	66% (3)	83% (3)	Blood. 1998 [29]
NHL-BFM90	BFM	1990–1995	K1/2 arm: 3/6 courses (MTX, DEX, oxazaphorins, ETP, AraC, DXR, IT) K3 arm: 6 intensified courses including HD-MTX/HD-AraC/HD-ETP	89	76% (5)	ND	Blood. 2001 [30]
NHL 9000/9602	UKCCSG	1990–1998	NHL 9000 including VCR, DXR, PSL, MTX, AraC, 6-TG NHL 9602 including COPADM × 3 with CYM (CY, MTX) × 2 intensified courses including HD-MTX/HD-AraC for CNS-positive disease	72	59% (5)	65% (5)	Br. J. Haematol. 2002 [31]
LNH92	AIEOP	1993–1997	Induction therapy (CY, VCR, DEX, DNR, IT), consolidation therapy (6-TG, AraC, ASP, HD-MTX, IT) with maintenance treatment	34	65% (5)	85% (5)	Cancer. 2005 [32]
POG9315	POG	1994–2000	APO (DXR, VCR, PSL, 6MP, MTX) with randomization of ID-MTX and HD-AraC	86	72% (4)	88% (4)	J. Clin. Oncol. 2005 [33]
CCG5941	CCG	1996–2001	Induction therapy (VCR, PSL, CY, DNR, ASP, IT, G-CSF), consolidation therapy (VCR, PSL, ETP, 6-TG, AraC, ASP, MTX, IT, G-CSF) with maintenance treatment	86	68% (5)	80% (5)	Pediatr. Blood Cancer. 2009 [34]
ALCL99	EICNHL	1999–2006	DEX, CY, IT, IFO, AraC, ETP with randomization of VBL	352	72% (10)	92% (10)	Cancers (Basel). 2020 [18]
ANHL0131	COG	2004–2008	APO (DXR, VCR, PSL, 6MP, MTX) with randomization of VBL	125	Non-VBL arm: 74% (3) VBL arm: 79% (3)	Non-VBL arm: 84% (3) VBL arm: 86% (3)	Pediatr. Blood Cancer. 2014 [35]

ALK, anaplastic lymphoma kinase; ALCL, anaplastic large cell lymphoma; EFS, event-free survival rates; OS, overall survival; SFOP, French Society of Pediatric Oncology; BFM, Berlin-Frankfurt-Munster; UKCCSG, United Kingdom Children’s Cancer Study Group; AIEOP, Italian Association of Pediatric Hematology and Oncology; POG, Pediatric Oncology Group; CCG, Children’s Cancer Study Group; CY, cyclophosphamide; DXR, doxorubicin; PSL, prednisolone; MTX, methotrexate; VCR, vincristine; DEX, dexamethasone; ETP, etoposide; AraC, cytarabine; IT, intrathecal therapy; HD, high-dose; 6-TG, 6-thioguanine; DNR, daunorubicin; ASP, asparaginase; 6-MP, 6-mercaptopurine; ID, intermediate-dose; IFO, ifosphamide; VBL, vinblastine; ND, no data; CNS, central nervous system.

**Table 2 cancers-15-05733-t002:** Characteristics of salvage therapies for ALK-positive ALCL.

		Number of Patients	CR Rate	EFS (Year)	OS (Year)	CNS Penetrability	Adverse Effect	Reference
Weekly vinblastine		31	83%	65% (5)	30% (5)	Low	Mild leukopenia (34%)	J. Pediatr. 2001 [39], J. Clin. Oncol. 2009 [9]
Brentuximab vedotin		17	41% (ORR: 53%)	ND	ND	Low	Fever (44%), nausea (36%), peripheral neuropathy (33%)	Lancet Haematol. 2018 [11]
ALK inhibitor	Crizotinib	26	81% (ORR: 88%)	ND	ND	Low	Visual disturbance (40–60%), gastrointestinal symptoms (29–54%)	J. Clin. Oncol. 2017 [40]
	Alectinib	10	60% (ORR: 80%)	70% (1)	70% (1)	High	Gastrointestinal symptoms (<15%)	Cancer Sci. 2020 [41]
	Ceritinib	3	2 out of 3 cases: CR 1 out of 3 cases: PR	ND	ND	ND	Gastrointestinal symptoms (32–82%)	Blood. 2015 [42], ESMO Open. 2021 [43]
Nivolumab		2 case reports	2 out of 2 cases: CR	ND	ND	ND	ND	Ann. Intern. Med. 2016 [44], Pediatr. Blood Cancer. 2018 [45]

ALK, anaplastic lymphoma kinase; ALCL, anaplastic large cell lymphoma; CR, complete response; EFS, event-free survival rates; OS, overall survival; ORR, overall response ratio; PR, partial response; HSCT, hematopoietic stem cell transplantation; ND, no data; CNS, central nervous system.

**Table 3 cancers-15-05733-t003:** Survival cases with ALK-positive ALCL with leukemic presentation.

Case	Age (Year)/Sex	CNS Lesion	Extranodal Lesion	1st Line	2nd Line	3rd Line	HSCT	Status at HSCT	Conditioning Regimen	Donor Source	Reference
1	1/M	ND	Li, Sp, PE	CsA, mPSL, CY, AraC, DXR, VP, MTX → CR			-	-			Blood. 2001 [74]
2	12/F	-	Li, Sp	ALCL99+VBL → CR			-	-			Br. J. Haematol. 2014 [23]
3	10/F	-	Li, Sp, Sk, Lu	ALCL99 → PR	DEX, VDS, AraC, VP → CR		+	1st CR	ND	ND	Br. J. Haematol. 2014 [23]
4	11/M	-	Li, Sp, Sk, Lu, A	ALCL99 → CR → relapse	VBL+glucocorticoid → 2nd CR		+	2nd CR	ND	ND	Br. J. Haematol. 2014 [23]
5	18/F	-	Li, Sp, Sk, Lu	CHOP → relapse	chemotherapy		+	2nd CR	ND	Allo	Br. J. Haematol. 1999 [72]
6	6/F	-	K, Lu	DXR, PSL, VCR → PR	MTX, IFO, VP, DEX → PR	AraC, CCNU, VBL, BLM → PR	+	PR	TBI/TEPA/VP/CY/alemtuzumab	Allo	Am. J. Clin. Pathol. 2003 [22]
7	40/M	+	Li, Sp	chemotherapy for ALL → PR, CNS+	IT, MTX, AraC → CNS-		+	PR	TBI/VP/CY	Allo	Int. Hematol. 2013 [20]
8	10/M	-	Li, Sp, Lu, Sk, PE	ALCL99 → PR, CNS+	alectinib → CR		+	1st CR	TBI/VP/CY	Allo	Br. J. Haematol. 2022 [21]

ALK, anaplastic lymphoma kinase; ALCL, anaplastic large cell lymphoma; CNS, central nervous system; HSCT, hematopoietic stem cell transplantation; ND, no data; Li, liver; Sp, spleen; Lu, lung; Sk, skin; K, kidney; PE, pleural effusion; A, ascites; CsA, cyclosporin A; mPSL, methylprednisolone; CY, cyclophosphamide; AraC, cytarabine; DXR, doxorubicin; VP, etoposide; MTX, methotrexate; ALCL99, ALCL99 protocol regimen; VBL, vinblastine; CHOP, CHOP regimen; VCR, vincristine; ALL, acute lymphoblastic leukemia; CR, complete response; PR, partial response; CNS+, CNS lesion-positive; CNS-, CNS lesion-negative; DEX, dexamethasone; VDS, vindesine; IFO, ifosphamide; IT, intrathecal injection; CCNU, lomustine; BLM, bleomycin; TBI, total-body irradiation; TEPA, thiotepa; allo, allogeneic.

## Data Availability

The data presented in this study are available in this article.

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
