# Peer review of "Strategy for Pediatric Patients with Relapsed or Refractory Anaplastic Lymphoma Kinase-Positive Anaplastic Large Cell Lymphoma: A Review"

_cancers, 2023, doi:10.3390/cancers15245733_

Round 1

Reviewer 1 Report

Comments and Suggestions for Authors

The authors reviewed the treatment options for pediatric relapsed/refractory ALL-positive ALCL. There is a review of pediatric ALCL, including relapse, published by Cancers in 2018 (Cancers 2018, 10, 99), and this manuscript lacks new findings compared to the previous review.

Several important articles are not cited, and it would greatly benefit readers if the authors described the "strategy" for determining which treatment option should be adopted based on the evidence for different types of relapse/refractory cases.

P2, line 52: 1. Introduction

Please verify the frequency of TPM3-ALK by referencing several reports on ALK variants.

P2, line 73: 2. Clinical features

The manuscript states that bone marrow involvement is 30%, but in ALCL99, it is 12%, which might be misleading and requires a more detailed explanation. MDD should be elaborated upon in more detail, including the prognostic significance of RT-PCR, RQ-PCR, and dPCR, as well as the methods for evaluating each of these techniques. The description that 50% of ALCLs show bone marrow involvement by PCR needs a more comprehensive explanation, including the PCR method, to avoid misinterpretation that 50% of ALCLs are MDD-positive.

P3, line 108: 4.1. 

The Morphological pattern section summarizes the characteristics of the leukemic presentation and its prognosis, and it may be preferable to discuss it in a separate section dedicated to “leukemic presentation”.

P5, line 172: 5.1 . Newly diagnosed ALK-positive ALCL

For ANHL12P1, the result of the ALCL99+crizotinib arm has been also published in the Journal of Clinical Oncology, and this should be mentioned.

P9: 5.2.4.1. Autologous HSCT

This manuscript did not mention the “ALCL-Relapse Trial (J Clin Oncol. 2020 Dec 1;38(34):3999-4009. ).” Moreover, there is some retrospective data on autologous HSCT for relapsed/refractory pediatric ALCL, including data from the US (CIBMTR), France (SFCE), Japan, and elsewhere. Based on these reports, the efficacy of autologous HSCT for relapsed/refractory ALCL should be clearly stated.

P9: 5.2.4.2. Allogeneic HSCT

As mentioned in the 2018 review, the efficacy of reduced-intensity conditioning should be discussed.

Other

While its application to routine practice is likely to remain limited, nivolumab should be mentioned as a potential new treatment option for relapsed/refractory cases.

Author Response

Comment #1: The authors reviewed the treatment options for pediatric relapsed/refractory ALL-positive ALCL. There is a review of pediatric ALCL, including relapse, published by Cancers in 2018(Cancers 2018, 10, 99), and this manuscript lacks new findings compared to the previous review. Several important articles are not cited, and it would greatly benefit readers if the authors described the "strategy" for determining which treatment option should be adopted based on the evidence for different types of relapse/refractory cases.

Answer to Comment #1: We appreciate your helpful suggestions on our manuscript. To address your comment, we have added a new section, “5.2.6 Treatment strategy for pediatric patients with relapsed or refractory ALK-positive ALCL,” at P14, Line 523–573, highlighted in red. This section suggests the “strategy” for determining which treatment option should be adopted for relapsed/refractory pediatric cases.

Comment #2: P2, line 52: 1. Introduction Please verify the frequency of TPM3-ALK by referencing several reports on ALK variants.

Answer to Comment #2: Thank you very much for your comment. We have added a reference to verify the frequency of TPM3-ALK at P2, Line 53.

Comment #3: P2, line 73: 2. Clinical features The manuscript states that bone marrow involvement is 30%, but in ALCL99, it is 12%, which might be misleading and requires a more detailed explanation. MDD should be elaborated upon in more detail, including the prognostic significance of RT-PCR, RQ-PCR, and dPCR, as well as the methods for evaluating each of these techniques. The description that 50% of ALCLs show bone marrow involvement by PCR needs a more comprehensive explanation, including the PCR method, to avoid misinterpretation that 50% of ALCLs are MDD-positive.

Answer to Comment #3: In virtue of your comment, we understand the necessity of a more detailed explanation of bone marrow involvement and MDD. We would like to explain bone marrow involvement in detail at P2, Line 72–76, highlighted in red. In addition, we would like to add a more detailed explanation about MDD at P5–6, Line 178–201, highlighted in red, and delete some words at P5, Line 174–177, P6, Line 191–195, 202-207, and P2, Line 76–80, highlighted in gray.

Comment #4: P3, line 108: 4.1. The Morphological pattern section summarizes the characteristics of the leukemic presentation and its prognosis, and it may be preferable to discuss it in a separate section dedicated to “leukemic presentation”.

Answer to Comment #4: We completely agree with your suggestion. Therefore, we would like to discuss the characteristics of the leukemic presentation in a separate section as “4.2. leukemic presentation” at P4–5, Line 143–170, highlighted in red, and delete sentences regarding leukemic presentation at P3–4, Line 110–142, highlighted in gray. In addition, we have added “leukemic presentation” as a risk factor, at P1, Line 13 and 26, at P3, Line 98, and at P15, Line 578, highlighted in red. In response to adding some words in Simple Summary and Abstract, we would like to delete some words, at P1, Line 17–18 and 34–35, highlighted in gray. In response to the new section “4.2. leukemic presentation”, we have changed each section number in “Minimal disseminated disease” and “CNS involvement”, at Line 171 and 208, respectively, highlighted in red.

Comment #5: P5, line 172: 5.1 . Newly diagnosed ALK-positive ALCL For ANHL12P1, the result of the ALCL99+crizotinib arm has been also published in the Journal of Clinical Oncology, and this should be mentioned.

Answer to Comment #5: Thank you very much for your comment. In virtue of your comment, we understand the result of the ALCL99+crizotinib arm. Therefore, we would like to mention about the result of the ALCL99+crizotinib arm at P7, Line 240–246, highlighted in red, and delete a word at Line 241, highlighted in gray.

Comment #6: P9: 5.2.4.1. Autologous HSCT This manuscript did not mention the “ALCL-Relapse Trial (J Clin Oncol. 2020 Dec 1;38(34):3999-4009. ).” Moreover, there is some retrospective data on autologous HSCT for relapsed/refractory pediatric ALCL, including data from the US (CIBMTR), France (SFCE), Japan, and elsewhere. Based on these reports, the efficacy of autologous HSCT for relapsed/refractory ALCL should be clearly stated.

Answer to Comment #6: In virtue of your comment, we understand that autologous HSCT was less effective for patients with early relapse in the ALCL-Relapse Trial. We would like to mention about the result of the ALCL-Relapse trial and several retrospective data on autologous HSCT for relapsed/refractory pediatric ALCL at P12, Line 435–460, highlighted in red. In addition, we would like to add an explanation at P12, Line 469, highlighted in red, and delete some words at P12, Line 469 and 471–472, highlighted in gray.

Comment #7: P9: 5.2.4.2. Allogeneic HSCT As mentioned in the 2018 review, the efficacy of reduced-intensity conditioning should be discussed.

Answer to Comment #7: We completely agree with your suggestion. Therefore, we would like to discuss the efficacy of reduced-intensity conditioning, at P13, Line 477–483, highlighted in red.

Comment #8: While its application to routine practice is likely to remain limited, nivolumab should be mentioned as a potential new treatment option for relapsed/refractory cases.

Answer to Comment #8: We completely agree with your suggestion. Therefore, we would like to discuss nivolumab in a separate section as “5.2.4. Nivolumab” at P11–12, Line 421–429, highlighted in red. In addition, we have added nivolumab as a potential new treatment option, at P8, Line 262–263, and in Table 2, highlighted in red. In response to the new section “5.2.4. Nivolumab”, we changed each section number in “Hematopoietic stem cell transplantation” at P12, Line 430, highlighted in red.

Reviewer 2 Report

Comments and Suggestions for Authors

Noguchi and Ikawa present a well written review of this topic. Comments:

-    In the treatment section they should explain briefly what the regimens used in the various protocols actually were. I also suggest a table with regimen, numbers and results.

-    They do not clearly provide a recommendation for paediatric patients – the authors should make some attempt to do so

-    The authors should comment on the (lack of) role of up-front transplant – is there a role in high risk disease (i.e. CNS disease)?

-    Table 1 should be simplified and give accurate numbers of patients involved and any PFS/EFS if available

-    The BV section can be shortened by removing the sentences on mechanism of action as this is generally well known

Comments on the Quality of English Language

nil

Author Response

Comment #1: In the treatment section they should explain briefly what the regimens used in the various protocols actually were. I also suggest a table with regimen, numbers and results.

Answer to Comment #1: We completely agree with your suggestion. Therefore, we would like to add a table with several clinical trial regimens, numbers, and results as Table 1. at Line 229 of the 6th page in the text, highlighted in red. In response to the new “Table 1”, we changed each Table number in “Characteristics of salvage therapies for ALK-positive ALCL” and “Survival cases with ALK-positive ALCL with leukemic presentation”, at Line 265 and 501, respectively, highlighted in red. In addition, we would like to briefly explain the existence of several clinical trial regimens at Line 220–222 of the 6th page, highlighted in red, and delete some words at Line 222, highlighted in gray.

Comment #2: They do not clearly provide a recommendation for paediatric patients – the authors should make some attempt to do so

Answer to Comment #2: We completely agree with your suggestion. Therefore, we would like to add a new section, “5.2.6 Treatment strategy for pediatric patients with relapsed or refractory ALK-positive ALCL,” at Line 523–573 of the 14–15th page, highlighted in red. This section explains the recommendation for pediatric patients with relapsed or refractory ALK-positive ALCL.

Comment #3: The authors should comment on the (lack of) role of up-front transplant – is there a role in high risk disease (i.e. CNS disease)?

Answer to Comment #3: We completely agree with your suggestion. Therefore, we would like to discuss the lack of role of up-front transplant, at Line 472–476 of the 12–13th page, highlighted in red.

Comment #4: Table 1 should be simplified and give accurate numbers of patients involved and any PFS/EFS if available

Answer to Comment #4: We completely agree with your suggestion. Therefore, we would like to add the data regarding “numbers of patients”, “EFS”, and “OS” to Table 2* (which was “Table 1” in the first manuscript) at Line 265 of the 8th page, highlighted in red. In addition, we would like to delete the data regarding “number of treatment cases for ALK-positive ALCL” to simplify our manuscript.

Comment #5: The BV section can be shortened by removing the sentences on mechanism of action as this is generally well known

Answer to Comment #5: We completely agree with your suggestion. Therefore, we would like to delete some sentences at Line 286–291 of the 9th page in the text, highlighted in gray.

Reviewer 3 Report

Comments and Suggestions for Authors

The authors put together a nice review of ALCL including most of the newer treatments such as ALK inhibitors. This would be a positive addition to the literature. 

Minor edits: 

Style of text in table 1: Uncapitalized headers and formatting 

Table 2: Capitalize headers

Would soften language in 5.2.4 (line 350): “can be” curative treatment if used in consolidation after relapse. SCT is often no curative and requires Mtn therapy or additional modalities along with it. 

Line 358: "first chemotherapy” (used multiple places) is defined as re-induction chemotherapy following first relapse? Or using AutoSCT during CR1? Would clarify sentence. In the US, the common phrase is frontline therapy. So this may be geographically correct in Japan. 

380: would change to frontline chemotherapy as it is more broadly used term 

382: would rephrase so timeline matches: they received salvage therapy followed by SCT. (SCT is consolidating the CR) except for two of the pts. who received SCT in PR status then used SCT as salvage regimen. 

414: Would also encourage here, and other parts of text, to consider high dose (HD) MTX, HD AraC as CNS penetrating modalities. LP+IT chemo is not traditionally used since these other methods penetrate the CNS well. 

Also, consolidating with XRT for focal disease should be considered in refractive patients. this could go in the SCT section. 

Author Response

Comment #1: Style of text in table 1: Uncapitalized headers and formatting, Table 2: Capitalize headers

Answer to Comment #1: Thank you very much for your comments. We capitalized headers and formatting in Table 2* and 3* (which was “Table 1” and “Table 2” in the first manuscript, respectively).

Comment #2: Would soften language in 5.2.4 (line 350): “can be” curative treatment if used in consolidation after relapse. SCT is often no curative and requires Mtn therapy or additional modalities along with it.

Answer to Comment #2: We completely agree with your suggestion. Therefore, we would like to add some words at Line 431–432 of the 12th page in the text, highlighted in red, and delete the word at Line 431 of the 12th page in the text, highlighted in gray.

Comment #3: Line 358: "first chemotherapy” (used multiple places) is defined as re-induction chemotherapy following first relapse? Or using AutoSCT during CR1? Would clarify sentence. In the US, the common phrase is frontline therapy. So this may be geographically correct in Japan.

Answer to Comment #3: Thank you very much for your comment. We would like to define “first chemotherapy” at Line 442 and 446 of the 12th page in 5.2.5.1(Autologous HSCT section) as “frontline therapy”, highlighted in red, and delete some words at Line 442–443 and 446 of the 12th page, highlighted in gray.

Comment #4: 380: would change to frontline chemotherapy as it is more broadly used term

Answer to Comment #4: We changed “first-line chemotherapy”, highlighted in gray, to “frontline chemotherapy”, highlighted in red at Line 490 and 492 of the 13th page in 5.2.5.2 (Allogeneic HSCT section).

Comment #5: 382: would rephrase so timeline matches: they received salvage therapy followed by SCT. (SCT is consolidating the CR) except for two of the pts. who received SCT in PR status then used SCT as salvage regimen.

Answer to Comment #5: We completely agree with your suggestion. Therefore, we would like to describe the detailed timeline at Line 493–495 of the 13th page in 5.2.5.2 (Allogeneic HSCT section), highlighted in red, and delete some words at Line 492–493 of the 13th page, highlighted in gray.

Comment #6: 414: Would also encourage here, and other parts of text, to consider high dose (HD) MTX, HD AraC as CNS penetrating modalities. LP+IT chemo is not traditionally used since these other methods penetrate the CNS well.

Answer to Comment #6: Thank you very much for your comment. We would like to explain HD-MTX and HD-AraC as CNS penetrating modalities for patients with CNS involvement, at Line 566–573 and 581 of the 15th page, highlighted in red.

Comment #7: Also, consolidating with XRT for focal disease should be considered in refractive patients. this could go in the SCT section.

Answer to Comment #7: In virtue to your comment, we understand that XRT should be considered for focal disease. Therefore, we would like to add a sentence explaining XRT for refractory patients, at Line 521–523 of the 14th page, highlighted in red. In addition, we would like to add a sentence explaining XRT for focal disease in adult patients, at Line 248–249 of the 7th page, highlighted in red.

Round 2

Reviewer 2 Report

Comments and Suggestions for Authors

all issues have been addressed

Reviewer 3 Report

Comments and Suggestions for Authors

Agree with the additions/corrections.

Suggestion is for Tables: They are not neat and need improvement in formatting with adjustment in fonts but maybe this is done in editing. 

Also title should have capitalized letters for proper words and after colon